# Evaluation of Entomopathogenic Nematodes against Common Wireworm Species in Potato Cultivation

**DOI:** 10.3390/pathogens12020288

**Published:** 2023-02-09

**Authors:** Arife Gümüş Askar, Ebubekir Yüksel, Refik Bozbuğa, Atilla Öcal, Halil Kütük, Dilek Dinçer, Ramazan Canhilal, Abdelfattah A. Dababat, Mustafa İmren

**Affiliations:** 1Istanbul Directorate of Agricultural Quarantine, Bakırköy, 34149 Istanbul, Türkiye; 2Department of Plant Protection, Faculty of Agriculture, Kayseri Erciyes University, Melikgazi, 38030 Kayseri, Türkiye; 3Department of Plant Protection, Faculty of Agriculture, Eskişehir Osmangazi University, Odunpazarı, 26160 Eskişehir, Türkiye; 4Atatürk Horticultural Central Research Institute, Merkez, 77100 Yalova, Türkiye; 5Department of Plant Protection, Faculty of Agriculture, Bolu Abant Izzet Baysal University, Gölköy, 14030 Bolu, Türkiye; 6Biological Control Research Institute, Yüreğir, 01321 Adana, Türkiye; 7International Maize and Wheat Improvement Centre (CIMMYT) 39, Emek, 06511 Ankara, Türkiye

**Keywords:** Elateridae, potato, beneficial nematodes, biocontrol, *Agriotes* spp., *Steinernema* spp., *Heterorhabditis* spp.

## Abstract

Wireworms (Coleoptera: Elateridae) are common insect pests that attack a wide range of economically important crops including potatoes. The control of wireworms is of prime importance in potato production due to the potential damage of the larvae to tuber quantity and quality. Chemical insecticides, the main control strategy against wireworms, generally fail to provide satisfactory control due to the lack of available chemicals and the soil-dwelling habits of the larvae. In the last decades, new eco-friendly concepts have emerged in the sustainable control of wireworms, one of which is entomopathogenic nematodes (EPNs). EPNs are soil-inhabitant organisms and represent an ecological approach to controlling a great variety of soil-dwelling insect pests. In this study, the susceptibility of *Agriotes sputator* Linnaeus and *A. rufipalpis* Brullé larvae, the most common wireworm species in potato cultivation in Türkiye, to native EPN strains [*Steinernema carpocapsae* (Sc_BL22), *S. feltiae* (Sf_BL24 and Sf_KAY4), and *Heterorhabditis bacteriophora* (Hb_KAY10 and Hb_AF12)] were evaluated at two temperatures (25 and 30 °C) in pot experiments. *Heterorhabditis bacteriophora* Hb_AF12 was the most effective strain at 30 °C six days post-inoculation and caused 37.5% mortality to *A. rufipalpis* larvae. *Agriotes sputator* larvae were more susceptible to tested EPNs at the same exposure time, and 50% mortality was achieved by two EPNs species, Hb_AF12 and Sc_BL22. All EPN species/strains induced mortality over 70% to both wireworm species at both temperatures at 100 IJs/cm^2^, 18 days post-treatment. The results suggest that tested EPN species/strains have great potential in the control of *A. sputator* and *A. rufipalpis* larvae.

## 1. Introduction

Wireworms, *Agriotes* spp. (Coleoptera: Elateridae), are one of the most destructive soil-dwelling insect pests that can cause severe economic losses in many crops including potato [1,2]. Wireworms generally live under soil during their larval development and feed on underground parts of plants such as seeds, roots, and tubers, leading to severe yield and tuber quality losses [3]. In addition to direct damage to tubers, the feeding holes of wireworm larvae on tubers predispose plants to subsequent secondary infections by other invertebrates and microbial pathogens [4]. Wireworm damage leads frequently to a drop in market value due to the visual quality of potatoes. In the absence of appropriate control measures, nearly half of the production can be downgraded in terms of quality [5]. The average price reductions due only to wireworm damage were estimated at 12% [4]. Therefore, potato crops are at greater risk for wireworm damage and require particular attention because of the damage potential of wireworms that render the tubers unmarketable [6]. Up to now, more than 39 wireworm species have been reported to attack potatoes around the world [7]. However, 9 species, *Agriotes brevis* Candèze, *Agriotes lineatus* Linnaeus, *Agriotes litigiosus* Rossi, *Agriotes obscurus* Linnaeus, *Agriotes proximus* Linnaeus, *Agriotes rufipalpis* Brullé, *Agriotes sordidus* Illiger, *Agriotes sputator* Linnaeus, and *Agriotes ustulatus* Schäller, are considered the most devastating in Europe [8,9].

The control of wireworms is quite challenging due to their extensive and hidden life cycle in the soil, and non-uniform distribution of the larvae in the fields [10]. For many years, chemical insecticides with broad-spectrum active ingredients such as organochlorine, organophosphates, and carbamates have been the main strategy to control wireworms by most producers. However, restrictions and bans in recent decades on the use of many of these synthetic chemicals, due to environmental and health concerns, have led many researchers to search for eco-friendly alternatives to synthetic insecticides for the management of wireworm populations [11,12,13,14]. Among new ecologically based approaches, entomopathogenic nematodes (EPNs) (Steinernematidae and Heterorhabditidae) have recently attracted a great deal of attention due to their biocontrol potential against many economically important insect pests [15,16,17]. Being soil-originated organisms, EPNs are natural suppressors of insect pest populations in the soil environment [18,19]. The infective juveniles (IJs) of *Steinernema* (Steinernematidae) and *Heterorhabditis* (Heterorhabditidae) species, stress-resistant stage surrounded by a protective sheath, seek out a potential host in soil without feeding and initiate the infection process by penetrating their hosts via body openings (mouth, anus, or spiracles) or cuticle [20,21]. Following penetration, the IJs move toward the host hemocoel and release their endosymbiotic bacteria (*Xenorhabdus* spp. and *Photorhabdus* spp., respectively) into the host hemolymph which induces toxemia and kills the host within 24–48 h [22,23]. The IJs feed on the bacterial cells and degraded host tissues through several generations until the depletion of food sources and then leave the host dead body to search for a new potential host [20].

The soil-dwelling characteristics and an efficient host-searching mechanism of EPNs make them a perfect candidate for biological control of soil-borne insect pests such as wireworms [24]. However, the success of EPNs against any kind of insect pest is heavily dependent on the adaptation capability of EPNs in the application area and matching the most appropriate EPN species/strains with the target pests [25]. In general, local EPNs are considered well-adapted to climatic and environmental conditions where they are isolated and can effectively suppress the pest populations without adverse effects on non-target species [26,27,28,29]. Therefore, in a previous study, a wide-ranging field survey was conducted in major potato cultivation areas of Türkiye to identify EPN species for the control of major potato pests [30]. In the present study, the effectiveness of isolated EPN species/strains was evaluated against the larvae of *A. sputator* and *A. rufipalpis* which are the two predominant wireworm species in potato growing areas of Türkiye [31].

## 2. Materials and Methods

### 2.1. Source of Nematodes

Three EPN species that were previously recovered from potato fields were employed in in vitro bioassays [30] (Table 1). In order to obtain a fresh batch of IJs, EPN species were multiplied in vitro on the last instar larvae of *Galleria mellonella* (L.) (Lepidoptera: Pyralidae) [25]. The IJs were suspended in 1 mL of sterile water at a concentration 200 IJs/Petri dish and inoculated to Petri plates (Ø9 cm) with autoclaved sandy loamy soil (20 g). The Petri plates were covered with parafilm and maintained at 25 °C and 65% relative humidity (RH). Dead Galleria larvae were collected daily with soft forceps and placed into modified white traps [32]. The harvested IJs were washed several times with sterile water and stored at 15 °C horizontally in cell culture flasks (250 mL) until bioassays were performed. The initial culture of *G. mellonella* larvae was obtained from the Entomology Laboratory of Erciyes University and reared on an artificial diet as described by Metwally et al. [33]. The larvae were reared in glass wide-neck jars (1 L) which were sterilized by autoclaving. The diet consisted of the following ingredients: wheat flour, wheat bran, milk powder, maize flour, dried yeast powder, honey, and glycerin. Approximately 100 1st instar larvae were put into each jar and the jars were maintained under laboratory conditions (30 °C and 65% relative humidity). The diet was refreshed every 20 days until the last instar larvae were obtained [33].

### 2.2. Source of Wireworms

*Agriotes rufipalpis* and *A. sputator* larvae were assembled from potato fields in different parts of Türkiye and identified based on the mitochondrial cytochrome c oxidase subunit I (COI) sequences in a previous study [31]. The collected larvae were brought individually to the laboratory in plastic containers (50 mL) containing autoclaved sandy loamy soil (20 g) with a slice of potato (Ø2 cm) and kept at 25 °C. The 4th and 5th larvae were separated according to their head width and observed for one week for any sign of infection [9,34,35]. Only healthy larvae were included in the pathogenicity bioassays. The pathogenicity of EPNs was tested against the mixed groups of 4th and 5th larval instars.

### 2.3. Evaluation of the Pathogenicity of EPNs in Pot Experiment

The effectiveness of EPNs was evaluated at two temperatures (25 and 30 °C) in pots (1.5 L) (Surface area 240 cm^2^) including 1 kg autoclaved (121 °C for 60 min) sandy loamy soil and a slice of fresh potato (Ø2 cm). The soil used in our experiment was obtained from Bolu Abant İzzet Baysal University (Department of Seed Science and Technology) and consisted of 81% sand, 14% silt, and 5% clay. The organic matter content of the soil was 2.1% with a pH of 6.5. Prior to the inoculation of IJs, each pot was irrigated with 100 mL of distilled water to provide an adequate amount of moisture for IJs. One larva of 4th or 5th instar was placed into pots and allowed to move deeper in the soil profile. The IJs suspended in 5 mL distilled water were applied uniformly to the soil surface at the concentration of 25, 50, 100, and 150 IJs/cm^2^ with the help of an automatic pipette (corresponding to 6000, 12,000, 24,000, and 36,000 IJs per pot, respectively). Then, the pots were maintained at 25 °C and 30 °C, 65% RH. Since moisture is one of the most important factors affecting the movement of IJs, the pots were irrigated with 50 mL of distilled water daily during the experiment. The larval mortality was checked at 6 and 18 days after treatment. Dead larvae that were transferred to white traps were observed under a stereomicroscope to confirm the nematode infection. There were four replicates of each treatment with ten larvae per replicate. All experiments were repeated twice on different dates.

### 2.4. Statistical Analysis

No mortality occurred in the control groups. Data pooled from two experiments were analyzed using IBM SPSS Statistics, Version 20.0 for Windows (SPSS Inc., Chicago, IL, USA). Prior to analyses, a normality test was performed and data were subjected to arcsine transformation. To determine significant differences among treatments, full-factorial model repeated-measures ANOVA was applied. The effects of the main factors (Nematode, Temperature, and IJs concentrations) and their interactions were considered significant at α = 0.05. Post-hoc comparisons were performed using Tukey’s multiple comparison test (*p* ≤ 0.05).

## 3. Results

The results revealed that the larvae of *A. rufipalpis* and *A. sputator* showed varying degrees of susceptibility to all tested EPN species and strains (Figure 1). All main factors had a significant effect on the mortality rates of both wireworm species (Table 2). Increasing concentrations of IJs and exposure time generally led to higher mortalities in the *A. rufipalpis* larvae, and mortality rates ranged between 2.5 and 37.5% at both temperatures tested. The susceptibility of *A. rufipalpis* larvae to EPNs tended to increase at 30 °C, and three EPN species/strains induced mortality over 30% at 6 days after treatment (DAT). *Heterorhabditis bacterophora* AF12 was the most efficient strain at 6 DAT and yielded 37.5% mortality at the highest concentration (150 IJs/cm^2^) (Table 3).

At the lowest concentration, only two *H. bacteriophora* isolates (Hb_KAY10 and Hb_AF12) were able to cause mortalities of over 40% at 18 DAT. A remarkable increase occurred in the mortality rates of *A. rufipalpis* larvae at 100 and 150 IJs/cm^2^ concentrations at 30 °C, and all EPN species/strains induced mortality of over 80% (Table 4). All EPN species/strains were able to kill the larvae of *A. sputator* at tested IJ concentrations and temperatures except for *S. feltiae* BL24 and *H. bacteriophora* KAY10 which induced no mortality at 25 IJs/cm^2^ at 25 °C. However, with increasing IJ concentrations, *H. bacteriophora* KAY10 yielded the highest mortality (47.5%) at 100 and 150 Ijs/cm^2^ at 25 °C and 6 DAT followed by *H. bacteriophora* AF12 (45.0%) and *S. feltiae* KAY4 (42.5%). Although *S. feltiae* BL22 was the least efficient isolate at 25 °C, the efficiency of BL22 significantly increased at 30 °C and achieved the highest mortality (50.0%) at 6 DAT along with *H. bacteriophora* KAY10 (Table 5).

After 18 days of exposure to the Ijs, the larval mortality substantially increased and all *H. bacteriophora* strains caused mortality over 50% at the lowest concentration (25 Ijs/cm^2^) at 25 °C. At 100 Ijs/cm^2^ concentrations, KAY10 and N3 strains of *H. bacteriophora* performed better than other EPN species/strains and induced 80 and 82.5% mortality, respectively. All EPN species/strains did not differ significantly at 100 and 150 Ijs/cm^2^ concentrations at 30 °C, and mortality ranged between 82.5 and 87.5% at 18 DAT. KAY10 and AF12 strains of *H. bacteriophora* were the only EPN species/strains that cause mortality over 65% at 25 Ijs/cm^2^ concentrations at 30 °C at 18 DAT. The highest efficacy (87.5%) was obtained from *H. bacteriophora* AF12 strain at 150 Ijs/cm^2^ at 18 DAT (Table 6).

## 4. Discussion

In the present study, native EPN species were tested for biocontrol potential against the most abundantly collected wireworm species from potato fields which is a major concern for potato growers in Europe including Türkiye [31,32,33,34,35,36]. As compared to other host species of EPNs, wireworms generally exhibited lower susceptibility to nematode infection, and mortality generally occurred over a longer period of time [37]. For instance, Williams et al. [38] tested the efficacy of *S. feltiae*, *S. carpocapsae*, *H. bacteriophora*, and *H. indica* on the larvae of *Melanotus communis* (G.) at 100 Ijs/cm^2^ concentration and the highest larval mortality did not exceed 15%. In another study, Forgia et al. [39] conducted a laboratory bioassay against *Agriotes sordidus* (Illiger) larvae in well plates at 2 Ijs/cm^2^ concentration and reported 8.3 and 16.7% of mortality for *S. carpocapsae* and *H. bacteriophora* strains, respectively. Campos-Herrera and Gutiérrez [40] evaluated the pathogenicity of different strains of *S. feltiae* and *S. carpocapsae* in well plates at 250 Ijs/cm^2^ and reported 9% mortality in larvae of *A. sordidus* 12 DAT which was induced by only one *S. feltiae* strain. In contrast to aforementioned studies, Toba et al. [41] reported 58% mortality in the 7–10th instars larvae of *Limonius californicus* (M.) when *S. feltiae* was applied at a concentration of 393 Ijs/cm^2^. In another study, Morton and Garcia-del-Pino [42] conducted a laboratory bioassay with different *S. feltiae*, *S. carpocapsae*, and *H. bacteriophora* strains against the 5th/6th larvae of *A. obscurus*, and mortality rates ranged between 17% and 35% for *H. bacteriophora* strains while the highest mortality (75%) was obtained with *S. carpocapsae* B14 strain at a concentration of 100 IJs/cm^2^. In the present study, all tested EPN species/strains were highly pathogenic to *A. rufipalpis* and *A. sputator* larvae at 18 DAT, and mortality over 70% was achieved by all EPN species/strains. The high virulence of tested EPNs in this study is in line with Ansari et al. [34], Morton and Garcia-del-Pino [42], and Sandhi et al. [43] who reported mortality between 50 and 75% in wireworm species. It is a well-known fact that the pathogenicity of EPNs on the same host species differs greatly among species and even strains [44,45]. Therefore, preliminary pathogenicity screening tests provide valuable information before evaluation of performance of EPNs [46]. In the present study, the most pathogenic EPN species that proved to be highly virulent on the larvae of *G. mellonella* in an earlier study were used [30]. In the aforementioned studies, different wireworm and EPN species were utilized in the bioassays which might be one possible reason leading to variations in the mortality rates. Earlier studies illustrated species-dependent nematode infection among a range of hosts [47,48]. Unsuitable EPN species-host combinations may be partially responsible for the differences in the mortality rates of wireworm larvae. In the current study, all EPN species and strains that were collected from potato fields infested with *A. sputator* and *A. rufipalpis* were used in the bioassays. It is reasonable to assume that tested EPN species/strains were predisposed to successfully infect and kill wireworm larvae. The co-existence of the tested EPNs within the host habitat may have helped to give additional positive responses to wireworm-derived cues and eventually to explain the high infection rates [47,49]. On the other hand, infection of EPNs requires successful penetration of IJs into the host body. Wireworms are considered to have strong morphological structures that help them to avoid or limit the penetration of IJs [50]. The differences in the morphological structures of wireworm species that function as physical barriers to EPNs might be another reason behind differences in the mortality rates. In addition, the immune systems of host species that detect the presence of microbial infection play a key role in the pathogenicity of IJs. Rahatkhah et al. [51] indicated that there could be a great variation in the recognition of IJs by the immune system of different wireworm species. Conversely, EPNs and their bacterial associates produce a large number of metabolites in the host hemocoel that exhibit immunosuppressant activity with varying levels of efficiency as well as toxicity to the host intestine [52,53,54]. A great variation in the chemical composition of secondary metabolites produced by different species and strains of *Xenorhabdus* and *Photohabdus* bacteria was also reported in earlier studies which may have contributed to the differences in the mortality of wireworm larvae [55,56]. Furthermore, the developmental stage of the host insect is one of the key factors affecting the pathogenicity of EPNs. Williams et al. [38] reported that smaller *M. communis* larvae showed higher susceptibility to EPNs compared to larger wireworm larvae. Morton and Garcia-del-Pino [42] reported 75% mortality against the 5th/6th instars larvae of *A. obscurus* which is in line with our study. The immune responses of insects show variation among developmental stages and late instars larvae may have more immune responses against pathogens [57,58]. Ebssa and Koppenhöfer [59] reported that the 4th and 5th instars of *Agrotis ipsilon* (H.) (Lepidoptera: Noctuidae) were the most susceptible stages to EPN species. In another study, Abdolmaleki et al. [58] stated that the 4th instar larvae of *Pieris brassicae* (L.) (Lepidoptera: Pieridae) demonstrated higher immune responses to bacterial associates of EPNs *Photorhabdus temperata* subsp. *temperate* than the 3rd instars. In the aforementioned studies, the efficacy of EPNs was tested against mixed instars of wireworm species and this might have also affected the effectiveness of EPNs. In addition, considering the number of larval instars of wireworm species (up to 13 instars), the 4th/5th instars larvae of *A. sputator* and *A. rufipalpis* may be the most susceptible stages against tested EPNs [35].

Environmental variables also have an influence on the effectiveness of EPNs, and the adaptation capability of EPNs to environmental factors varies greatly among species and strains [60,61]. Temperature and humidity are among the major environmental factors that can either enhance or diminish the survival, mobility, and virulence of EPNs [62,63,64]. In the present study, a significant increase was observed in the efficacy of *H. bacteriophora* strains, particularly at low concentrations at 18 DAT. In previous studies, the optimum temperatures were reported to range between 22 and 24 °C for *S. feltiae* and 14 and 35 °C for *S. carpocapsae*, while *H. bacteriophora* reported to perform better between 25 and 30 °C [65,66]. The higher performance of *H. bacteriophora* strains at low concentrations may be explained by the optimal temperatures needed by the strains to successfully infect their host.

## 5. Conclusions

In this study, all tested EPN species/strains were highly pathogenic to *A. rufipalpis* and *A. sputator* larvae, and mortality over 70% was achieved at 18 DAT by all EPN species/strains. The results obtained indicated that EPN species that were recovered from the potato fields where EPNs and wireworms co-exist have the potential to provide better control against wireworms. However, field evaluation of these EPN species and strains will provide better insights into the performance of EPNs. In field conditions, several wireworm species in different development stages may be present with varying susceptibility to EPNs. Therefore, combining several EPN species may help suppress the wireworm populations.

## Figures and Tables

**Figure 1 pathogens-12-00288-f001:**
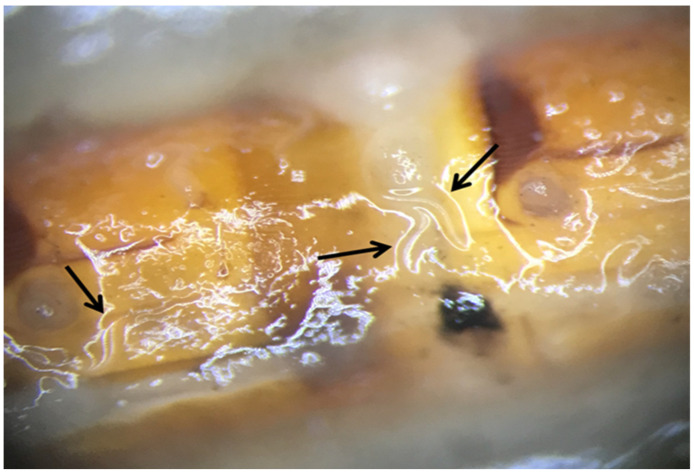
Emerging infective juveniles (black arrows) of *Steinernema feltiae* from the body of *Agriotes rufipalpis* larvae.

**Table 1 pathogens-12-00288-t001:** List of entomopathogenic nematode species/strains used in the experiments.

Entomopathogenic Nematodes	Strain	Habitat	Coordinates	GenBank Accession Number
*Steinernema carpocapsae*	Sc_BL22	Potato	40°47′11″ N 31°38′78″ E	OK632299
*Steinernema feltiae*	Sf_BL24	Potato	40°47′14″ N 31°39′10″ E	OK632300
*Steinernema feltiae*	Sf_KAY4	Potato	38°20′28″ N 35°27′49″ E	OK632306
*Heterorhabditis bacteriophora*	Hb_N3	Potato	38°02′19″ N 34°44′18″ E	OK632328
*Heterorhabditis bacteriophora*	Hb_KAY10	Potato	38°16′49″ N 35°25′15″ E	OK632308
*Heterorhabditis bacteriophora*	Hb_AF12	Potato	37°55′31″ N 29°52′19″ E	OK632288

**Table 2 pathogens-12-00288-t002:** Repeated-measures analysis of variance parameters for the main factors and associated interactions (Tukey, *p* ≤ 0.05).

Sources		*Agriotes rufipalpis*		*Agriotes sputator*	
	Degree of Freedom	F Value	*p* Value	F Value	*p* Value
Nematode (N)	5	9.268	<0.01	17.455	<0.01
Concentration (C)	5	295.432	<0.01	825.869	<0.01
Temperature (T)	1	42.688	<0.01	57.436	<0.01
C × N	25	1.735	0.020	3.327	<0.01
C × T	5	15.012	<0.01	5.465	<0.01
N × T	5	2.746	0.020	2.042	0.074
C × N × T	25	1.554	0.051	4.895	<0.01
Error1	216				
Exposure time (t)	1	491.683	<0.01	337.559	<0.01
t × C	5	94.844	<0.01	28.474	<0.01
t × N	5	1.225	0.298	2.961	0.013
t × T	1	77.778	0.540	3.194	0.075
t × C × N	25	0.799	0.742	1.457	0.081
t × C × T	5	5.321	0.358	1.636	0.152
t × N × T	5	5.054	0.011	3.206	0.008
t × C × N × T	25	1.717	0.022	1.340	0.137
Error2	216				

**Table 3 pathogens-12-00288-t003:** Mortality rates (%) of 4th/5th instars larvae of *Agriotes rufipalpis* 6 days after application of different entomopathogenic nematode species/strains in the pot experiments.

Temperatures	Nematodes *	Mortality Rates (%) 6 Days after Treatment (DAT)				
		Control	25 IJs/cm^2^	50 IJs/cm^2^	100 IJs/cm^2^	150 IJs/cm^2^
25 °C	Sc_BL22	0.0 ± 0.0A ^a^ a ^b^	7.5 ± 5.0Ba	7.5 ± 5.0Ba	10.0 ± 8.1Ba	12.5 ± 5.0Ba
	Sf_BL24	0.0 ± 0.0Aa	2.5 ± 5.0Aa	5.0 ± 5.7Aa	15.0 ± 5.7Ba	25.0 ± 5.7Cb
	Sf_KAY4	0.0 ± 0.0Aa	2.5 ± 5.0Aa	5.0 ± 5.7Aa	25.0 ± 5.7Bb	25.0 ± 5.7Bb
	Hb_N3	0.0 ± 0.0Aa	2.5 ± 5.0Aa	2.5 ± 5.0Aa	15.0 ± 10.0Ba	27.5 ± 9.5Cb
	Hb_KAY10	0.0 ± 0.0Aa	5.0 ± 5.7Aa	5.0 ± 10.0Aa	15.0 ± 10.0Ba	20.0 ± 8.1Bab
	Hb_AF12	0.0 ± 0.0Aa	7.5 ± 5.0Ba	7.5 ± 9.5Ba	15.0 ± 5.7Ca	15.0 ± 5.7Ca
30 °C	Sc_BL22	0.0 ± 0.0Aa	12.5 ± 5.0Ba	12.5 ± 9.5Ba	15.0 ± 8.1Ba	22.5 ± 5.0Ca
	Sf_BL24	0.0 ± 0.0Aa	7.5 ± 9.5Ba	17.5 ± 5.0Cab	30.0 ± 8.1Db	32.5 ± 5.0Dab
	Sf_KAY4	0.0 ± 0.0Aa	15.0 ± 5.7Ba	25.0 ± 5.7Cb	27.5 ± 5.0Cb	32.5 ± 5.0Dab
	Hb_N3	0.0 ± 0.0Aa	2.5 ± 5.0Aa	10.0 ± 11.5Ba	25.0 ± 12.9Cb	27.5 ± 9.5Ca
	Hb_KAY10	0.0 ± 0.0Aa	10.0 ± 0.0Ba	10.0 ± 0.0Ba	12.5 ± 5.0Ba	25.0 ± 10.0Ca
	Hb_AF12	0.0 ± 0.0Aa	15.0 ± 10.0Ba	20.0 ± 0.0Bb	25.0 ± 5.7Bb	37.5 ± 5.0Cb

* Sc_BL22: *Steinernema carpocapsae*; Sf_BL24 and Sf_KAY4: *Steinernema feltiae*; Hb_N3, Hb_KAY10, and Hb_AF12: *Heterorhabditis bacteriophora*. ^a^ Different capital letters show statistically significant differences among the infective juvenile concentrations (IJs) for each entomopathogenic nematode species. ^b^ Different lowercase letters show statistically significant differences among entomopathogenic nematode species/strains for each infective juvenile concentration (*p* < 0.05, Tukey).

**Table 4 pathogens-12-00288-t004:** Mortality rates (%) of 4th/5th instars larvae of *Agriotes rufipalpis* 18 days after application of different entomopathogenic nematode species/strains in the pot experiments.

Temperatures	Nematodes *	Mortality Rates (%) 18 Days after Treatment (DAT)				
		Control	25 Ijs/cm^2^	50 Ijs/cm^2^	100 Ijs/cm^2^	150 Ijs/cm^2^
25 °C	Sc_BL22	0.0 ± 0.0A ^a^ a ^b^	30.0 ± 0.0Bb	45.0 ± 10.0Bab	82.5 ± 5.0Cba	85.0 ± 11.5Cb
	Sf_BL24	0.0 ± 0.0Aa	25.0 ± 5.7Bab	37.5 ± 5.0Ba	77.5 ± 5.0Ca	85.0 ± 12.9Cb
	Sf_KAY4	0.0 ± 0.0Aa	25.0 ± 5.7Bab	37.5 ± 5.0Ba	80.5 ± 15.0Ca	85.0 ± 5.7Cb
	Hb_N3	0.0 ± 0.0Aa	25.0 ± 5.7Bab	55.5 ± 5.0Cb	77.5 ± 5.0Da	80.0 ± 8.1Da
	Hb_KAY10	0.0 ± 0.0Aa	20.0 ± 8.1Ba	60.0 ± 8.1Cb	80.0 ± 5.7Ca	80.5 ± 12.5Da
	Hb_AF12	0.0 ± 0.0Aa	25.0 ± 5.7Bab	50.0 ± 0.0Cab	72.5 ± 5.0Da	85.0 ± 12.9Db
30 °C	Sc_BL22	0.0 ± 0.0Aa	30.0 ± 0.0Ba	55.0 ± 10.0Ca	82.5 ± 5.0Bda	85.0 ± 11.5Da
	Sf_BL24	0.0 ± 0.0Aa	35.0 ± 5.7Aa	67.5 ± 5.0Bab	87.5 ± 5.0Ca	87.5 ± 10.5Ca
	Sf_KAY4	0.0 ± 0.0Aa	25.0 ± 5.7Aa	67.5 ± 5.0Bab	82.5 ± 15.0Ca	85.0 ± 5.7Ca
	Hb_N3	0.0 ± 0.0Aa	35.0 ± 5.7Ba	65.5 ± 5.0Bab	80.5 ± 5.0Ca	80.0 ± 8.1Ca
	Hb_KAY10	0.0 ± 0.0Aa	40.0 ± 8.1Cab	75.0 ± 8.1Db	80.0 ± 5.7Da	82.5 ± 12.5Da
	Hb_AF12	0.0 ± 0.0Aa	45.0 ± 5.7Cb	75.0 ± 0.0Db	85.5 ± 5.0Da	85.0 ± 12.9Da

* Sc_BL22: *Steinernema carpocapsae*; Sf_BL24 and Sf_KAY4: *S. feltiae*; Hb_N3, Hb_KAY10, and Hb_AF12: *Heterorhabditis bacteriophora*. ^a^ Different capital letters show statistically significant differences among the infective juvenile concentrations (Ijs) for each entomopathogenic nematode species. ^b^ Different lowercase letters show statistically significant differences among entomopathogenic nematode species/strains for each infective juvenile concentration (*p* < 0.05, Tukey).

**Table 5 pathogens-12-00288-t005:** Mortality rates (%) of 4th/5th instars larvae of *Agriotes sputator* 6 days after application of different entomopathogenic nematode species/strains in the pot experiments.

Temperatures	Nematodes *	Mortality Rates (%) 6 Days after Treatment (DAT)				
		Control	25 Ijs/cm^2^	50 Ijs/cm^2^	100 Ijs/cm^2^	150 Ijs/cm^2^
25 °C	Sc_BL22	0.0 ± 0.0A ^a^ a ^b^	5.0 ± 5.7Aa	10.0 ± 0.0Aba	12.5 ± 9.5Aba	17.5 ± 5.0Ba
	Sf_BL24	0.0 ± 0.0Aa	0.0 ± 0.0Aa	7.5 ± 9.5Ba	27.5 ± 5.0Cab	27.5 ± 5.0Cab
	Sf_KAY4	0.0 ± 0.0Aa	2.5 ± 5.0Aa	5.0 ± 5.7Aa	25.0 ± 5.7Bab	42.5 ± 5.0Cb
	Hb_N3	0.0 ± 0.0Aa	7.5 ± 9.5Ba	17.5 ± 5.0Cab	20.0 ± 0.0Cab	37.5 ± 5.0Db
	Hb_KAY10	0.0 ± 0.0Aa	0.0 ± 0.0Aa	12.5 ± 9.5Bab	40.0 ± 8.1Cb	47.5 ± 9.5Cb
	Hb_AF12	0.0 ± 0.0Aa	7.5 ± 9.5Ba	15.0 ± 5.7BCb	32.5 ± 5.0Cab	45.0 ± 5.7Db
30 °C	Sc_BL22	0.0 ± 0.0Aa	5.0 ± 5.7Aa	15.0 ± 5.7Ba	32.5 ± 5.0Cb	50.0 ± 8.1Db
	Sf_BL24	0.0 ± 0.0Aa	10.0 ± 11.5Ba	22.5 ± 9.5Ca	25.0 ± 12.9Cab	37.5 ± 5.0Da
	Sf_KAY4	0.0 ± 0.0Aa	10.0 ± 8.1Ba	20.0 ± 0.0Ca	32.5 ± 5.0Db	40.0 ± 8.1Da
	Hb_N3	0.0 ± 0.0Aa	7.5 ± 9.5Ba	17.5 ± 5.0Ca	20.0 ± 0.0Ca	37.5 ± 5.0Da
	Hb_KAY10	0.0 ± 0.0Aa	15.0 ± 10.0Ba	20.0 ± 8.1Ba	35.0 ± 5.7Cb	50.0 ± 8.1Db
	Hb_AF12	0.0 ± 0.0Aa	15.0 ± 5.7Ba	22.5 ± 5.0Ba	27.5 ± 5.0BCb	40.0 ± 8.1Ca

* Sc_BL22: *Steinernema carpocapsae*; Sf_BL24 and Sf_KAY4: *S. feltiae*; Hb_N3, Hb_KAY10, and Hb_AF12: *Heterorhabditis bacteriophora*. ^a^ Different capital letters show statistically significant differences among the infective juvenile concentrations (Ijs) for each entomopathogenic nematode species. ^b^ Different lowercase letters show statistically significant differences among entomopathogenic nematode species/strains for each infective juvenile concentration (*p* < 0.05, Tukey).

**Table 6 pathogens-12-00288-t006:** Mortality rates (%) of 4th/5th instars larvae of *Agriotes sputator* 18 days after application of different entomopathogenic nematode species/strains in the pot experiments.

Temperatures	Nematodes *	Mortality Rates (%) 18 Days after Treatment (DAT)				
		Control	25 Ijs/cm^2^	50 Ijs/cm^2^	100 Ijs/cm^2^	150 Ijs/cm^2^
25 °C	Sc_BL22	0.0 ± 0.0A ^a^ a ^b^	45.0 ± 5.7Ba	55.0 ± 5.7Ba	75.0 ± 5.7Ca	85.0 ± 5.7Ca
	Sf_BL24	0.0 ± 0.0Aa	35.0 ± 5.7Ba	65.0 ± 5.7Cb	75.0 ± 5.7Ca	75.0 ± 5.7Ca
	Sf_KAY4	0.0 ± 0.0Aa	42.5 ± 5.0Ca	52.5 ± 5.0Ca	72.5 ± 5.0Da	72.5 ± 5.0Da
	Hb_N3	0.0 ± 0.0Aa	52.5 ± 5.0Cab	62.5 ± 5.0Ca	82.5 ± 5.0Da	82.5 ± 5.0Da
	Hb_KAY10	0.0 ± 0.0Aa	60.0 ± 11.5Cb	70.0 ± 11.5CDb	80.0 ± 11.5Da	85.0 ± 11.5Da
	Hb_AF12	0.0 ± 0.0Aa	57.5 ± 5.0Cab	77.5 ± 5.0Dc	77.5 ± 5.0Da	80.0 ± 5.0Da
30 °C	Sc_BL22	0.0 ± 0.0Aa	55.0 ± 5.7Bca	75.0 ± 5.7Ca	85.0 ± 5.7Ca	85.0 ± 5.7Ca
	Sf_BL24	0.0 ± 0.0Aa	55.0 ± 5.7Ca	75.0 ± 5.7Da	85.0 ± 5.7Da	85.0 ± 5.7Da
	Sf_KAY4	0.0 ± 0.0Aa	52.5 ± 5.0Ca	72.5 ± 5.0Da	82.5 ± 5.0Da	82.5 ± 5.0Da
	Hb_N3	0.0 ± 0.0Aa	55.0 ± 5.0Ca	80.0 ± 5.0Da	82.5 ± 5.0Da	82.5 ± 5.0Da
	Hb_KAY10	0.0 ± 0.0Aa	65.0 ± 11.5Cb	80.0 ± 11.5Da	85.0 ± 11.5Da	85.0 ± 11.5Da
	Hb_AF12	0.0 ± 0.0Aa	67.5 ± 5.0Bb	80.0 ± 5.0Ca	82.5 ± 5.0Ca	87.5 ± 5.0Ca

* Sc_BL22: *Steinernema carpocapsae*; Sf_BL24 and Sf_KAY4: *S. feltiae*; Hb_N3, Hb_KAY10, and Hb_AF12: *Heterorhabditis bacteriophora*. ^a^ Different capital letters show statistically significant differences among the infective juvenile concentrations (Ijs) for each entomopathogenic nematode species. ^b^ Different lowercase letters show statistically significant differences among entomopathogenic nematode species/strains for each infective juvenile concentration (*p* < 0.05, Tukey).

## Data Availability

Data generated in this study are available upon reasonable request to the corresponding author.

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
