# Peer review of "Evaluation of Entomopathogenic Nematodes against Common Wireworm Species in Potato Cultivation"

_pathogens, 2023, doi:10.3390/pathogens12020288_

Round 1

Reviewer 1 Report

The article titled " Evaluation of Entomopathogenic Nematodes against Common Wireworm Species in Potato Cultivation" found that EPN species that were recovered from the potato fields where EPNs and wireworms co-exist have the potential to provide better control against wireworms. This study is interesting and provides evidence for pest control. However, following shortcomings need to be overcome:

Introduction

1.     Line 46,“Agriotes” should be italicized.

2.     Line 60-61, the full name of a species should be used when it first appears.

Results

1.     Table 3 and 5, For the same nematode species, there appears to be no difference between the control treatment (mortality is 0) and the other concentration treatments, such as 25 IJs/cm2 and 50 IJs/cm2. Especially the first row of data in Table 5. It does not seem correct. Please check.

2.     Table 4, the first row of data. I can see capital letters A and C used to show significant differences, but letter B is not found. It should be marked in alphabetical order. The current marking method seems to be wrong. Please revise.

3.     Line 189, “Agriotes rufipalpis” should be italicized

4.     Line 198, “Agriotes sputator” should be italicized

References:
1. Some references have doi links, most do not. Please be consistent.

2.Line 343, Missing comma between “2022” and “13”.

3. Line 396-398, I'm not sure if unpublished articles can be cited in the references.

Author Response

Dear reviewers,

Thank you for your valuable comments on the revised manuscript. The manuscript has been re-revised according to your suggestions and comments. All the corrections made were colored in yellow in the manuscript.

Reviewer #1:

  • Line 46,“Agriotes” should be italicized.
  • Line 189, “Agriotes rufipalpis” should be italicized
  • Line 198, “Agriotes sputator” should be italicized

  • Dear reviewer, thank you for carefully reviewing the manuscript.
    Agriotes, Agriotes rufipalpis, and Agriotes sputator was italicized.

  • (2) Line 60-61, the full name of a species should be used when it first appears.
  • Dear reviewer, thank you for carefully reviewing the manuscript. All the scientific names in the text were corrected following your suggestions.

  • Table 3 and 5, For the same nematode species, there appears to be no difference between the control treatment (mortality is 0) and the other concentration treatments, such as 25 IJs/cm2 and 50 IJs/cm2. Especially the first row of data in Table 5. It does not seem correct. Please check.
  • Dear reviewer, thank you for carefully reviewing the manuscript. Tables were checked and necessary corrections were made.

  • Table 4, the first row of data. I can see capital letters A and C used to show significant differences, but letter B is not found. It should be marked in alphabetical order. The current marking method seems to be wrong. Please revise.
  • Dear reviewer, thank you for carefully reviewing the manuscript. The table was revised following your comments.

  • References:
  1. Some references have doi links, most do not. Please be consistent.
  2. Line 343, Missing comma between “2022” and “13”.
  3. Line 396-398, I'm not sure if unpublished articles can be cited in the references.

  • Dear reviewer, thank you for carefully reviewing the manuscript. The references were corrected. According to the MDPI reference, unpublished articles can be cited in the references.

Author 1, A.B.; Author 2, C. Title of Unpublished Work. Abbreviated Journal Name year, phrase indicating stage of publication (submitted; accepted; in press).

Reviewer 2 Report

My comments and advices for authors are made in the text.

Some parts should be better explained and more clear, there are italics missing, mistakes on the tables subtitles, etc.

The references should be carefully reviewed according to the journal rules. Some are not.

Author Response

Dear reviewers,

Thank you for your valuable comments on the revised manuscript. The manuscript has been re-revised according to your suggestions and comments. All the corrections made were colored in yellow in the manuscript.

Reviewer #2: Thank you for your valuable comments on the revised manuscript. The manuscript has been re-revised according to your suggestions and comments. All the corrections made were colored in yellow in the manuscript.

  • Some parts should be better explained and more clear, there are italics missing, mistakes on the tables subtitles, etc.
  • Dear reviewer, thank you for carefully reviewing the manuscript. A paragraph was added to the Materials and Methods section following your suggestions as follows:
    The larvae were reared in glass wide-neck jars (1 L) which were sterilized by autoclaving. The diet consisted of the following ingredients: wheat flour, wheat bran, milk powder, maize flour, dried yeast powder, honey, and glycerin. Approximately 100 1st instar larvae were put into each jar and the jars were maintained under laboratory conditions (30 ºC and 65% relative humidity). The diet was refreshed every 20 days until the last instar larvae were obtained [33].

  • The references should be carefully reviewed according to the journal rules. Some are not.

  • Dear reviewer, thank you for carefully reviewing the manuscript. The references were checked and corrected.